# Empirical Compression Model of Ultra-High-Performance Concrete Considering the Effect of Cement Hydration on Particle Packing Characteristics

**DOI:** 10.3390/ma16134585

**Published:** 2023-06-25

**Authors:** Pengfei Li, Xiaoyan Wang, Hanbo Cao

**Affiliations:** Department of Harbor, Waterway, and Coastal Engineering, Chongqing Jiaotong University, Chongqing 400074, China; 622200090056@mails.cqjtu.edu.cn (X.W.); 622200090012@mails.cqjtu.edu.cn (H.C.)

**Keywords:** UHPC, multi-component hydration, microscopic pore development, strength prediction method

## Abstract

The mix design of UHPC has always been based on a large number of experiments; in order to reduce the number of repeated experiments, in this study, silica fume (SF), fly ash (FA), and limestone powder (LP) were used as the raw materials to conduct 15 groups of experiments to determine the particle size distribution (PSD) properties of UHPC. A model of multi-component hydration based on the SF, FA, and LP pozzolanic reactions was devised to quantify the rate and total heat release during the hydration process. Additionally, a microscopic pore development model, which was based on the accumulation of hydration products, was established to measure the effect of these products on the particle-packing properties. Utilizing this model, a UHPC strength prediction technique was formulated to precisely forecast the compressive strength based on a restricted experimental data set. The applicability of this prediction method was verified using 15 sets of existing experimental data along with the data collected from 4 research articles. The results show that the prediction method can predict the strength values of different mix proportions with an accuracy rate of over 80%.

## 1. Introduction

Ultra-high-performance concrete (UHPC), a form of concrete designed to maximize the particle size distribution (PSD) of cementitious materials, was founded on the notion of reactive powder concrete (RPC). To increase uniformity and reduce defects, ultrafine reactive powder is added and normally coarse aggregates are not included. To bolster toughness and reduce the water to binder and porosity ratios, steel fibers are also incorporated [1,2]. UHPC has very high strength, toughness, and durability compared with ordinary concrete [3]. A recent research project on UHPC, the hydraulic tunnel project in China [4,5], is considered to be of great importance. This project features many of the usual traits of UHPC, such as its remarkable depth of burial, intense in situ pressure, and intricate geological conditions. As an effective way to resolve relevant problems, the concrete of the hydraulic tunnel was released from water, and UHPC was developed for the tunnel lining structure. A few studies [4,5] have been conducted to investigate the comprehensive value of using UHPC for improving the service life of structures.

Considering that there are few theoretical bases for UHPC design, further experimental studies are essential for achieving UHPC [6]. A few mix design methods have been investigated to simplify the UHPC design workload [7,8,9,10]. Four types of mix design methods have been proposed in the literature, which are based on rheological properties, statistical designs, neural network models, and close packing models [11,12,13,14,15]. No theoretical model exists for the design of mixed materials based on rheological properties, thus necessitating the constant alteration of these properties. A considerable quantity of experimental data must be supplied to ensure the precision of a model; however, neither statistical design techniques nor neural network models have been employed in a combined design. Close packing models, which are categorized into discrete and continuous models, are the focus of this study. Composed of compressible and linear packing models, discrete models are highly intricate due to their reliance on the filling fraction of single solid components and these components’ combinations. However, the filling fraction of these fine materials is hard to ascertain in the mix design process [16]. The other type of model is the particle packing model with continuous grading, which is defined as a continuous model. The grade curve of maximum particle density, which was proposed by Fuller and Thomson [17,18,19,20,21,22], assumes the foundation of a continuous model. Furnas [18] assumed that particles do not interfere with each other and that PSD is continuous, which was expressed by the densification of small-particle systems. The modified Andreasen and Andersen (MAA) particle packing model can be used to construct UHPCs that are both dense and uniform [19]. Yu et al. [20] proposed a close packing model of particles for optimization of the composition and gradation of raw materials. The authors demonstrated that UHPC with a low cement content and high compressive strength could be conveniently designed using the MAA model. To date, the MAA model is the most widely used method in UHPC mix design due to its convenient use and feasibility.

The MAA model, which is essential for gauging dimensional trends, forecasts the strength of concrete based on the initial packing. Nevertheless, the initial packing condition is not able to accurately evaluate the concrete’s strength, as it is constantly influenced by the initial buildup of materials and hydration of the cement. To accurately forecast the strength of concrete, it is necessary to precisely assess the impact of hydration products on the concrete’s stacking properties during hydration. The hydration of cement involves the interaction between cement-based materials, hydration formation, and micro-pores development [23,24,25]. The rapid reaction of silica fume (SF), a raw material commonly used in UHPC, accelerates the hydration of cement with the calcium hydroxide that is produced during the process, thus significantly increasing the rate of hydration of tricalcium silicate (C3S) [26,27,28,29] and more hydrated calcium silicate (C-S-H). Apart from cement, the materials that are used are mainly SF, fly ash (FA), and limestone powder (LP). The use of these materials, in turn, densifies the fine structure of the cement substrate. LP’s increased fineness can hasten the hydration reaction of tricalcium silicate (C_3_S) [30,31,32] as it increases the number of nucleation sites, thus augmenting the rate of cement hydration in the initial stages. FA is an inert material in the early hydration stages because of its low pozzolanic activity. This material plays a filling role and does not react with other substances. The activity of certain fly ash materials [33] determines the reaction time. By replacing cement with FA, Liu [34] observed a delay in the production of hydration heat; the total heat release also lessened with an increase in FA content, which was likely due to the slow reaction of FA and the dilution effect of cement. Therefore, it is essential to examine two aspects when gauging the influence of the cement hydration reaction on the initial amassment and concrete strength: (1) ascertain the hydration magnitude of each element in various mixing proportions; and (2) gauge the effect of hydration on the dispersal of hydration products and initial aggregation. Maekawa [35,36] conducted a study on the influence of various powders on the hydration exothermic reaction; the author proposed a multi-component hydration model for composite cementitious materials that was based on Jiang’s [37] model, which examined the time-dependent retardation effect of epoxy latexes on cement hydration. The results indicated the model’s great potential for predicting the kinetics and thermodynamics of cement hydration with varying epoxy latex contents.

The aim of this research was to construct an empirical model for gauging the com-pressive strength of various concrete mix ratios. Experiments of varying mix ratios were conducted to achieve this, and their compressive strength values were gauged. Using the resulting data, an empirical model was developed that could accurately predict the compressive strength of a given mix ratio based on its constituent materials. The model was able to accurately forecast the compressive strength of various mix ratios, thereby demonstrating its wide-ranging applicability and practicality in improving concrete performance. These results have a considerable effect on the design and construction of concrete structures, providing a reliable means for predicting the compressive strength of different mix ratios and ensuring their longevity and dependability.

## 2. Materials and Methods

The experimental design is described in this section. First, the initial states and hydration paths of the cement used in this study were obtained. In this study, P.O.52.5 Portland ordinary cement, FA, LP, and SF were used as the raw cementitious materials.

The chemical and mineralogical makeup of Portland cement is detailed in Table 1, which was obtained using an X-ray fluorescence spectrum analysis (XRF) and the Bogue method. The Brunauer–Emmett–Teller (BET) method was used to determine the specific surface area of the powders, with FA displaying a specific surface area of 465 m^2^/kg and an apparent density of 2.55 g/cm^3^ and LP demonstrating a specific surface area and apparent density of 600 m^2^/kg and 2.84 g/cm^3^, respectively. SF had an S_i_O_2_ content of >95%, a specific surface area of 2000 m^2^/kg, and an apparent density of 2.25 g/cm^3^.

Two kinds of sand were used: coarse quartz sand with a size of 0–4.75 mm and fine quartz sand with the size of 0–2 mm. The density of these two kinds of sand was 2.64 g/cm^3^. To enhance the structure’s mechanical properties, straight steel fibers that were 0.2 mm in diameter and 13 mm in length were employed. To adjust the workability of the UHPC, a poly-carboxylic acid-based superplasticizer (SP) was employed with a dosage of 30 kg/m^3^ for the mix design.

### 2.1. Mix Design

Table 2 presents 15 amalgamations of cement-based substances, with the water to binder and sand to binder ratios being 0.20 and 1.1, respectively. These substances are cementitious and comprise cement, SF, FA, and LP.

### 2.2. Preparation of UHPC and Curing

All mixtures were created and examined at a temperature of 20 ± 2 °C. The lab conditions necessitated us to perform the mixing of dried aggregates and powder materials.

The mixing was performed in three steps:

(1)Put all dried powder materials and sand in a mixer for 60 s to mix.(2)Add water and SP to 80%, remix the sample for 360 s, and then stop the mixer for 30 s.(3)Mix water, SP, and steel fibers together; add the remaining ingredients, and continue mixing for a further 240 s.

Cubes of 100 mm × 100 mm × 100 mm were utilized for the compressive strength testing. After casting, the specimens decomposed approximately 24 h later and were then exposed to customary curing conditions (temperature: 20 ± 2 °C; humidity: >99%). After 7 and 28 days of curing, the compressive strengths of the specimens were evaluated.

The GB/T 31387-2015 test for mechanical properties necessitates a cubic specimen with the dimensions of 100 mm × 100 mm × 100 mm for compressive strength; a loading rate ranging from 1.2 MPa/s to 1.4 MPa/s; and a prism specimen with the dimensions of 100 mm × 100 mm × 400 mm for flexural strength, with a loading rate ranging from 0.08 MPa/s to 0.1 MPa/s.

## 3. Experimental Results of Compressive Strength

### 3.1. Experimental Results

To examine the impact of diverse mineral blends on the compressive strength of UHPC concrete, SF, FA, and LP were chosen for the mix formation.

The results are presented in Table 3. The compressive strength of UHPC increases with the increase in the concentration of mineral amalgamations after 7 and 28 days of curing. As the SF content in the system rises to 10% and 40%, the compressive strength of the UHPC increases by 7% and 38%, respectively; this increase is mainly attributed to SF’ s dense microstructure and rapid pozzolanic reaction rate [38]. After seven days of curing, the compressive strength of UHPC with SF being replaced by cement is greater than that of UHPC with FA or LP. Nevertheless, when FA replaces cement, the average 7-day compressive strength of UHPC decreases by 4% compared with UHPC with SF, implying that FA’s pozzolanic reaction is sluggish and incomplete. After 28 days, the pozzolanic reaction intensifies, thus diminishing the disparity between UHPC’s compressive strength when FA or SF is substituted for cement [39]. However, LP, being an inert admixture, only plays a filling role in the cement paste [40]. No discernible effect of UHPC’s density on its potency is visible, and its compressive strength is inferior to that of SF and FA.

When the content of SF is 10%, its 28-day compressive strength meets the requirement of 100 MPa (Table 3). From an economic perspective, the experiments that were conducted with multi-powder material instead of cement used a 10% SF content along with a minimum compressive strength of 100 MPa, as per the GB/T 31387-2015 standard. Thus, FA or LP was used instead of cement while keeping the SF content at 10%. As shown in Table 3, there is a gradual rise in compressive strength over the 7-day and 28-day periods with the rise in FA content. As the FA content rises from 10% to 20%, the compressive strength of the system increases; however, when the FA content surpasses 20%, it decreases. A slight rise in compressive strength is also seen with the increase in LP content.

### 3.2. Prediction of Strength Trend Based on the MAA Model

Based on the experiments (Section 2.1, Section 2.2 and Section 3.1), the MAA model was used to predict the strength trend of concrete with different combinations of cement-based material. The formula for the MAA model is as follows:(1)$$P(D_{i})=\frac{D_{i}^{q}-D_{\min}^{q}}{D_{\max}^{q}-D_{\min}^{q}}$$
where the percentage of particle size *D* (%) is denoted by *P*(*D_i_*); the maximum particle size (μm) is *D_max_*; the minimum particle size is *D_min_*; and the distribution modulus, *q*, is determined by the ratio of large to small particles in the system. The larger the particles, the larger the modulus. Figure 1 illustrates the mixture design grading curves of 15 groups of selected powders with different proportions.

The least square method (LSM) was used to adjust the mass ratio of the raw materials in the dry mixture. The residual sum of squares (*RSS*) was used to further elucidate the divergence between the cumulative distribution curve of the mixture and the model target curve, which is derived from various amounts of raw materials. The RSS was calculated as follows:(2)$$RSS={\sum_{i=1}^{n}\left(P_{mix}(D_{i}^{i+1})-P_{tra}(D_{i}^{i+1})\right)}^{2}$$
where *RSS* is the sum of the squares of residuals and *Pmix* and *Ptra* are the actual accumulation curve and target curve, respectively.

The raw materials’ particle sizes, *D_max_* and *D_min_*, were 9500 μm and 0.503 μm, respectively, at their peak and minimum. The distribution modulus (*q*) was 0.23 to establish the objective function of the MAA model (Figure 1). The MAA model’s RSS, which is a dimensionless metric, quantitatively reflects the close packing degree of UHPC particles. A lower RSS implies a higher packing density; thus, RSS reflects the initial accumulation state of concrete with varying mix designs. The MAA model’s calculated RSS value curve was then mirrored (Figure 2) to compare the compressive strength trend as predicted by the model to the actual test trend.

The results suggest that the initial accumulation state can predict future strength trend laterally and that the initial accumulation can qualitatively characterize the compressive strength of concrete. Therefore, this study only needed to consider the influence of hydration products based on the initial accumulation, which could accurately predict the compressive strength.

## 4. Multi-Component Hydration Model Considering the Influence of Different Powders

An examination of the influence of the raw materials on the compressive strength of UHPC was conducted, which was based on physical accumulation while disregarding the effect of alterations in the water to binder ratio on concrete performance. Thus, incorporating the chemical hydration reaction process into the accumulation model was essential. The hydration of cement and the effect of product volume alterations on accumulation are the primary discussion of the subsequent two sections. Figure 3 illustrates the entire process, including both the physical accumulation and the chemical hydration reaction.

### 4.1. Hydration of Cement in Concrete

The four major minerals of Portland cement, alite (C_3_S), belite (C_2_S), aluminate (mainly C_3_A), and ferrite (C_4_AF), are powdered cement clinker and gypsum (Figure 4). However, when cement clinker is burned, aluminate and ferrite (collectively known as the gap phase) fill the gaps between calcium silicates [35,36].

The proportions of mineral components in various types of cement vary. Therefore, a hydrothermal model suitable for any given kind of cement must accurately describe the heat dissipation process of cement based on its mineral content. Various pozzolans can be employed to reduce or substitute Portland cement, thereby inhibiting the production of heat. Therefore, different raw materials are regarded as one unit, and the hydration transfer process of different components must be defined based on their interactions. A hydrothermal model based on such multi-component concepts represents various types of cement based on their clinker mineral composition [40].

An original multi-component cement hydration model was developed by Koichi Maekawa [40], which calculates the hydrothermally generated rate of cement by using the sum of the individual hydrothermal heat rate of each component based on the percentage of each component; meanwhile, the exothermic reaction of each mineral is described separately. The total heat consumption of cement that contains the mixed powders (*H_C_*) is the sum of the heat consumption of the component reactions [35].
(3)$$Q_{t}=\int H_{c}dt$$
(4)$$\begin{array}{l} H_{c}=\sum P_{i}H_{i}=P_{C_{3}A}\left(H_{C_{3}AET}+H_{C_{3}A}\right)+P_{C_{4}AF}\left(H_{C_{4}AFET}+H_{C_{4}AF}\right)\\ +P_{C_{3}S}H_{C_{3}S}+P_{C_{2}S}H_{C_{2}S}+P_{SF}H_{SF}+P_{FA}H_{FA}+P_{LP}H_{LP} \end{array}$$
where *P_i_* is the weight composition ratio and *H_C3AET_
*and *H_C4AFET_* are the rates of heat generation during the formation of ettringite based on the aluminate and ferrite phases, respectively. After the reaction of calcium hydroxide formation has stopped due to the disappearance of unreacted gypsum, hydration heat is generated in C_3_A and C_4_AF (expressed as *H_C3A_* and *H_C4AF_*, respectively). *H_i_* is the heat release rate of mineral *i* per unit weight, as defined in Equation (5):(5)$$H_{i}=\gamma_{i}\cdot \beta_{i}\cdot \lambda_{i}\cdot \mu_{i}\cdot H_{i,T_{0}}\left(Q_{i}\right)\exp\left\{-\frac{E_{i}}{R}\right.\left.\left(\frac{1}{T}-\frac{1}{T_{0}}\right)\right\}$$
where the activation energy of component *i*, which is denoted by *E_i_*, is accompanied by the gas constant *R*, and the reference heat-generation rate of component *i* at a constant temperature *T_0_* is *H_i,T0_*. The coefficient of change when no other effect is present, *γ_i_*, is defined as the delaying effect of the chemical admixture and FA in the initial exothermic hydration process; *β_i_* is the decrease in the heat generation rate due to the reduction in the availability of free water. *λ_i_* is the coefficient of the heating rate change caused by the absence of Ca(OH)_2_ in the powder mixture’s liquid phase. A coefficient of heat generation rate alteration, *μ_i_*, is the interdependence between alite and belite in Portland cement. (−*E_i_*/*R*) is defined as the thermal activity.

### 4.2. Effect of Fly Ash on Hydration

FAs are typically regarded as single reaction units that blend cement. However, these components are considered in this model as a single component, and the mixing of cement partially suppresses the generation of heat. The reaction of the powder mixture is determined based on the hydration of cement, which is linked to the amount of Ca(OH)_2_ [36].
(6)$$H_{FA}=\min\left\{\left.\beta_{i}\lambda_{i}\right\}\right.H_{FA,T_{0}}\left(Q_{i}\right)\exp\left\{-\frac{E_{i}}{R}\right.\left.\left(\frac{1}{T}-\frac{1}{T_{0}}\right)\right\}$$
(7)$$H_{FA,T_{0}}\left(Q_{i}\right)=0.005\cdot {\left(1-\frac{Q}{Q_{\max}}\right)}^{2}\left(1.2\frac{s}{s_{0}}-0.2\right)$$
(8)$$\beta_{i}=1-\exp\left\{\left.-r\left\{{\left.\left(\frac{\omega_{free}}{100\eta_{i}}\right)/s_{i}^{0.5}\right\}}^{s}\right.\right\}\right.$$
where *r* and *s* are the material constants that are common to all minerals. The comparison of the experimental and analytical results show that *R* = 5.0 and *S* = 2.4. The coefficient of *β_i_* varies from 0 to 1, and *s_i_* is a function of the normalized Blaine value, which represents the change in the reference heating rate due to powder fineness. $w_{free}$ is the free water ratio, and $\eta_{i}$ is the internal reaction layer of the thickness component *i*.

It is essential for the model to calculate the amount of Ca(OH)_2_ consumed by cement hydration, as other powder mixtures require it. The rate of reduction after mixing is expressed as the ratio of the remaining Ca(OH)_2_ to the amount required for the active reaction of FA.
(9)$$\lambda_{i}=1-\exp\left\{\left.-2.0{\left(\frac{F_{CH}}{R_{FACH}}\right)}^{1.5}\right\}\right.$$
where *F_CH_* is the amount of Ca(OH)_2_, that is produced by the hydration of C_3_S and C_2_S but not consumed by the C_4_AF reaction, and *R_FACH_* is the amount of Ca(OH)_2_ required for the reaction with FA when a sufficient amount of Ca(OH)_2_ is available. 

Because of limited test equipment, this study did not carry out an isothermal exothermic test, but instead used the data obtained from isothermal exothermic tests in other scholars’ articles to verify the model. In the study by He [33], isothermal calorimetry tests were conducted on UHPC samples with different levels of FA content to measure the heat dissipation rate during the hydration process. A comparison was drawn between the outcomes of the isothermal experiment and the numerical simulations in the present research, which utilized the original multi-component hydration model.

Figure 5 illustrates the contrast between the experimental and numerical simulation outcomes of the UHPC samples with varying FA content in terms of heat dissipation rate. The experimental results demonstrate that the inclusion of FA can significantly reduce the heat dissipation rate of the UHPC samples. The original multi-component hydration model’s simulation results demonstrate a strong correlation with the experimental data, thereby demonstrating its broad applicability to UHPC with varying levels of FA content.

### 4.3. Effect of Silica Fume on Hydration

As an active admixture, the exothermic process of SF is similar to that of FA; thus, the heat release rate formula of SF is as follows:(10)$$H_{SF}=\min\left\{\left.\beta_{i}\lambda_{i}\right\}\right.H_{SF,T_{0}}\left(Q_{i}\right)\exp\left\{-\frac{E_{i}}{R}\right.\left.\left(\frac{1}{T}-\frac{1}{T_{0}}\right)\right\}$$
(11)$$H_{SF,T_{0}}\left(Q_{i}\right)=0.005\cdot {\left(1-\frac{Q}{Q_{\max}}\right)}^{2}\left(1.2\frac{s}{s_{0}}-0.2\right)$$
(12)$$\beta_{i}=1-\exp\left\{\left.-r\left\{{\left.\left(\frac{\omega_{free}}{100\eta_{i}}\right)/s_{i}^{0.5}\right\}}^{s}\right.\right\}\right.$$
(13)$$\lambda_{i}=1-\exp\left\{\left.-2.0{\left(\frac{F_{CH}}{R_{FACH}}\right)}^{1.5}\right\}\right.$$

A comparison of SF concrete’s isothermal heat dissipation test results (Figure 6) with its numerical simulation results was performed to guarantee the broad applicability of the multi-component hydration model’s original design.

To demonstrate the broad applicability of Kadri’s [26] isothermal heat dissipation test for SF concrete and its capacity for accurately predicting SF concrete behavior, a comparison was made between the numerical simulation results from the original multi-component hydration model and the results of the isothermal heat dissipation test. A comparison between the experimental and numerical results, as depicted in Figure 6 and corroborated by the model, is essential for understanding and optimizing SF concrete’s performance, as it captures the intricate interplay between chemical reactions and physical processes during the initial hydration stages.

### 4.4. Effect of Limestone on Hydration

In the multi-component hydration model presented in this study, LP was included as a powder material. LP, although not actively engaged in the hydration process, has a critical role in optimizing the model’s performance. The packing density and porosity are both increased and reduced by its micro-filling effect on other active components [41,42], thus enhancing the mechanical and chemical properties of the system. Figure 7 illustrates the proposed mechanism by which LP contributes to the micro-filling effect of the model.

For P2 and P3:(14)$$Q_{ij}^{\prime }=Q_{ij}$$
(15)$$HS_{ij}^{\prime }=HS_{ij}\cdot \left(1+k_{H1}\cdot r_{s}\right)$$

For P4 and P5:(16)$$Q_{ij}^{\prime }=Q_{i\max}-\left(Q_{i\max}-Q_{ij}\right)\cdot \left(1+k_{Q}\cdot r_{s}\right)$$
(17)$$HS_{ij}^{\prime }=HS_{ij}\cdot \left(1+k_{H2}\cdot r_{s}\right)$$

For P6:(18)$$Q_{ij}^{\prime }=Q_{i\max}-\left(Q_{i\max}-Q_{ij}\right)\cdot \left(1+k_{Q}\cdot r_{s}\right)$$
(19)$$HS_{ij}^{\prime }=HS_{ij}\cdot \left(1+k_{H3}\cdot r_{s}\right)$$
(20)$$r_{s}=(p_{LP}\cdot B_{LP})/(p_{C}\cdot B_{C})$$
where *j* is the point number in the reference heat rate function; *Q_ij_* and *HS_ij_* are the heat rate and accumulated heat of component *i*, respectively; *Q_imax_* is the maximum heat generation; *p_LP_* and *p_C_* are the weight fractions; and *B_LP_* and *B_C_* are the unit weight surface areas of LP and Portland cement, respectively. The acceleration effect is expressed by the ratio of the surface areas of LP and cement, *r_s_*, which is taken as an indicator. The coefficients *k_H1_*, *k_H2_*, *k_H3_*, and *k_Q_* are multiplied by *r_s_* to represent the degree of contribution of LP.

The results of the isothermal exothermic test for LP concrete, as reported by Moon [43], were compared with the numerical simulation results obtained using the original multi-component hydration model (Figure 8). This comparison was made to demonstrate the broad applicability of the multi-component hydration model and its ability to accurately predict the behavior of various types of concrete, including those containing LP as an inert material. The comparison affirms that the model is able to precisely capture the exothermic heat discharged during the initial hydration period, which is essential for comprehending and optimizing the performance of LP concrete. 

### 4.5. Simulation of Multi-Component Powders on Hydration

In the preceding sections, the hydration of binary powder systems (excluding cement) was analyzed in detail to examine the heat release and chemical reactions that occur during the early stages of hydration. The analysis yielded significant revelations regarding the conduct of these systems, aiding us in constructing a more thorough comprehension of their efficacy. In this section, the analysis was extended to ternary powder systems by using the developed multi-component hydration model to verify the heat release during hydration. Thongsanitgarn [44] delved into the data that were generated from an investigation of ternary powder systems to gain further understanding of the behavior of these systems as well as any distinctions or resemblances between the binary powder systems under study. Comparing and contrasting the outcomes of these analyses allowed for a more profound comprehension of the elements that affect the hydration of powder systems and how they can be improved for better performance.

Based on the literature data, the isothermal exothermic test results were compared with the numerical simulation results of the multi-component hydration model in this study (Figure 9). The overall exothermic fitting is more accurate, which reflects the wide applicability of the multi-component hydration model.

## 5. Porosity Development Model

### 5.1. Microscopic Influence of Hydration Products

Considering that it is the solid phase rather than the porous phase that sustains strength development, the volume ratio of hydration products to the initial capillary space *D_hyd.out_* must be taken into account, and the volume of hydrates formed outside the primitive cement particles was used as an index describing strength development [36].
(21)$$D_{hyd.out}=\frac{V_{hyd.out}}{V_{cap.ini}}=\frac{V_{hyd.total}-V_{hyd.in}}{V_{cap.ini}}$$
where *D_hyd.out_* is the ratio of the initial capillary space to the space occupied by a large amount of external hydrates; *V_hyd.out_* is the volume of hydrates formed outside the primitive cement particles; *V_hyd.total_* is the total volume of hydration products formed inside the original cement particles; *V_hyd.in_* is equivalent to the reaction volume fraction of mineral compounds; and *V_cap.ini_* is the volume of the initial capillary space. The volume changes of various hydration inner and outer products are shown in Figure 10.

The Schiller model [36] shows that the volume of the initial capillary space *V_cap.ini_* is equivalent to the porosity without strength. Although Schiller determined the value by fitting the estimated value with the experimental results, in the model proposed in this study, the volume of the initial capillary space *V_cap.ini_* was calculated based on the water to binder ratio of the mixture as follows:(22)$$V_{cap.ini}=\frac{W/C\cdot \rho_{C}}{W/C\cdot \rho_{C}+1}$$
where *W/C* is the water to binder ratio and *ρ_c_* is the cement density.

The compressive strength is impacted by the magnitude of the hydration products created by cement particles. These particles are shaped like spheres, and the mineral elements of cement clinker react with free water during the hydration process to create products. These spheres are divided into the non-hydration inner core, original boundary, hydrated inner product, and hydrated outer product. As shown in Figure 11, a lower water to binder ratio results in a higher compressive strength, regardless of whether *D_hyd.out_* is the same, due to the smaller spacing between particles in the mixture with a reduced water to binder ratio.

### 5.2. Effect of Hydration Products on Packing Density

The change factor *k_i_* of the hydration products’ accumulation state was taken into account by considering two parameters: the alteration in the volume and the gap between cement particles. The *k_i_* is calculated as follows:(23)$$k_{i}=\frac{D_{hyd.out}}{\theta }$$
(24)$$K=k_{c}+\sum \phi_{i}\left(k_{i}-k_{c}\right)=k_{c}+\phi_{SF}\left(k_{SF}-k_{c}\right)+\phi_{FA}\left(k_{FA}-k_{c}\right)+\phi_{LP}\left(k_{LP}-k_{c}\right)$$
(25)$$\phi_{i}=\frac{m_{i}}{m_{C}+m_{SF}+m_{FA}+m_{LP}}$$
where *D_hyd.out_* is the ratio of the space occupied by a large amount of external hydrates to the initial capillary space; *θ* is the gap effect between cement particles; *k_i_* is the hydration factor of different powders *i*, including SF, FA, and LP; *φ_i_* is the mass ratio of powders in the mixture; and *m_c_* is the mass of different powders *i* (kg/m^3^). 

The volume of hydration products dynamically changes according to the hydration reaction; thus, the stacking state of the entire structure changes accordingly. An accurate prediction of concrete strength should be quantified by considering the influence of the change in hydration product volume on stacking.

### 5.3. Empirical Strength Prediction Model

A model of strength prediction based on alterations in the porosity and hydration product was proposed in a previous study [36]. The model equations are as follows:(26)$$f_{c}=f_{\infty }^{\prime }\left[1-\exp\left\{\left.-\alpha^{\prime }K^{\beta^{\prime }}\right\}\right.\right]$$
(27)$$f_{\infty }^{\prime }=A\cdot p_{C}\frac{p_{C_{3}S}}{p_{C_{3}S}+p_{C_{2}S}}+B\cdot p_{C}\frac{p_{C_{2}S}}{p_{C_{3}S}+p_{C_{2}S}}+C\cdot p_{SF}+D\cdot p_{FA}+E\cdot p_{LP}$$
where *f_∞_* is the ultimate strength (N/mm^2^); α′ and β′ are the material constants; and *P_c_*, *P_SF_*, *P_FA_*, and *P_LP_* represent the weight fractions of Portland cement, SF, FA, and LP, respectively. *A*, *B*, *C*, *D*, and *E*, are the material constants, indicating the contribution factor of each component on the strength development. This model has *A* at 120, *B* at 70, *C* at 210, *D* at 240, and *E* at 180, which represent the critical measure of concrete’s strength and the water to binder ratio while taking into consideration the alterations in hydration products. In this model, the strength prediction of UHPC is limited to 0.14–0.22.

The accuracy of the model was verified using 15 sets of strength analysis data, which were obtained through experiments using different mix proportion design combinations (Table 2).

By comparing the measured compressive strength values of different mix designs with the compressive strength values simulated by the empirical model, the experiment proves that the empirical model can successfully quantify the compressive strength values, as shown in Figure 12. The model can accurately predict the compressive strength of various mix proportions and has wide applicability and practicability in improving the performance of concrete.

### 5.4. Simplified Mix Design Process

To enable an accurate prediction of concrete strength with minimal experimentation, a strength prediction equation was developed based on the changes in the volume and heat rate of water. By establishing a parameter system using only a few experiments, it is possible to use the concrete strength to accurately predict and optimize the mix design process for improved performance. The resulting simplified mix design process, shown in Figure 13, provides a practical and efficient method for designing and constructing concrete structures with predictable and reliable strength characteristics. By combining our strength prediction equation with the simplified mix design process, a comprehensive framework is offered for optimizing the performance of concrete, ensuring its durability and stability over time. These results provide a dependable and effective method for forecasting and refining the concrete strength and efficiency, thereby significantly influencing the design and construction of concrete structures.

### 5.5. Validation

The effectiveness of the simplified mix design process was verified by means of 23 sets of experimental data from various papers [45,46,47,48]. The purpose of this verification was to demonstrate the precision and dependability of the simplified mix design process in predicting the concrete strength and optimizing the mix designs for enhanced performance. The verification results, as depicted in Figure 14, affirm the efficiency of the simplified mix design process in precisely forecasting the concrete strength while simultaneously optimizing the mix designs.

A simulation was conducted to evaluate the efficacy of the simplified mix design process with the aim of ascertaining the ideal mix proportion and forecasting the strength value of concrete. The simulation was designed to demonstrate the dependability and precision of the simplified mix design process in forecasting concrete strength and optimizing mix designs for enhanced performance. The simulation results, as shown in Figure 14, confirm the effectiveness of the simplified mix design process in accurately predicting concrete strength values while simultaneously optimizing mix proportions.

## 6. Conclusions 

In this study, SF, FA, and LP were used as raw materials to conduct 15 groups of experiments to determine the different original PSD properties of UHPC. A model of multi-component hydration, which was based on the pozzolanic reaction of SF, FA, and LP, was formulated to quantify both the rate and total heat release of the hydration process. Additionally, a microscopic pore development model based on the accumulation of hydration products was created to measure the effect of these products on the particle packing characteristics. A UHPC strength prediction model was designed to measure strength, which was based on the abovementioned model and only used data from a few experiments. The precision of the proposed model was confirmed by contrasting the empirical outcomes with existing experimental outcomes. The conclusions of this study and prospects are as follows:

(1) The compressive strength tests using different powders were carried out. The results show that SF can significantly improve the compressive strength of concrete, and FA is better than LP in terms of improving the strength. The MAA model is used to simulate the experimental results of compressive strength with different mix designs, which can predict the trend of compressive strength under different mix designs well.

(2) A model of hydration heat release, which was composed of multiple components, was formulated to quantitatively gauge the influence of SF, FA, and LP. Verification through experimentation affirmed that the simulation of the reaction rate and heat release during the hydration process was very accurate. 

(3) A micro-pore development model was developed based on the accumulation of hydration products. Estimates were made regarding the fluctuation in the volume of distinct hydration products and the dynamic alteration of the cement particle gap. Based on these results, a complete set of formulas was established to create the parameter system.

(4) The verification of the proposed strength prediction method was accomplished through the utilization of experimental data obtained from the literature. The results suggest that the proposed method can achieve accurate strength prediction by utilizing data from only a few experiments. 

(5) In this study, the proposed method could reduce the amount of practical engineering experiments and guide the mix design; however, the UHPC strength model in this paper is only suitable for compressive strength, and the test results of flexural strength have not been further analyzed. It is suggested that in a follow-up study, the prediction model of the bending strength of UHPC with different powders could be considered to better guide the mix design’s mechanical properties in UHPC.

## Figures and Tables

**Figure 1 materials-16-04585-f001:**
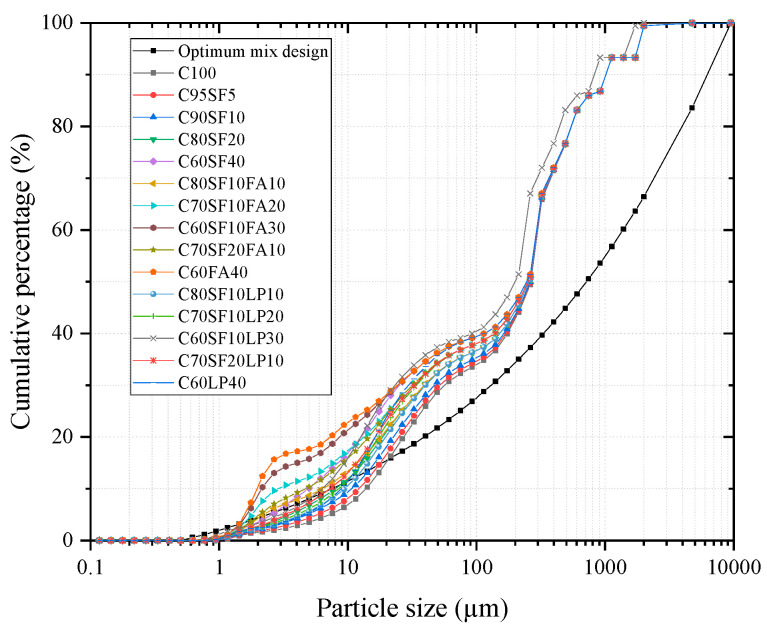
The mixing procedure.

**Figure 2 materials-16-04585-f002:**
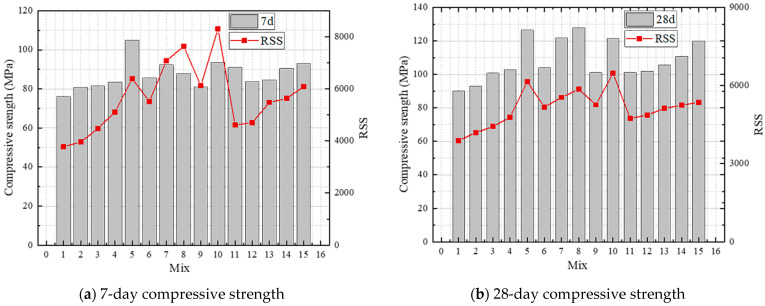
Validation of the MAA model in terms of compressive strength.

**Figure 3 materials-16-04585-f003:**
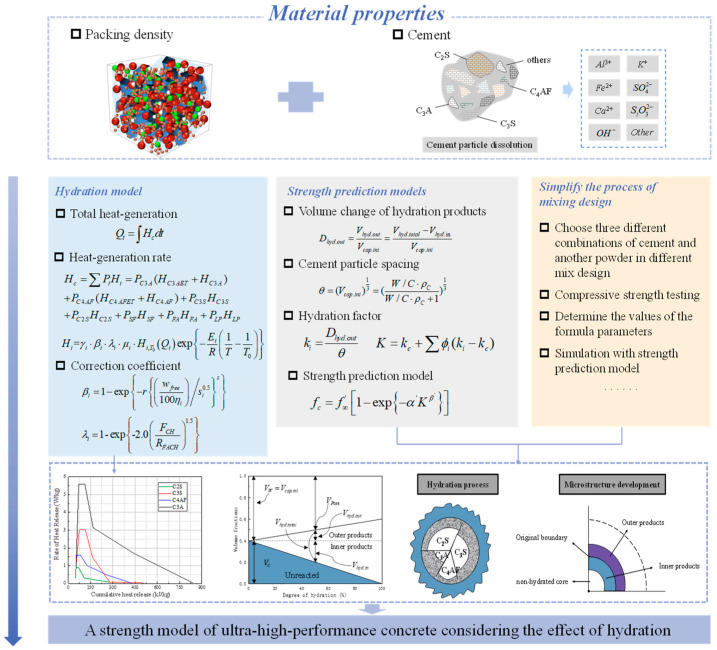
Research and development process of a compressive strength model of ultra-high-performance concrete considering hydration effect.

**Figure 4 materials-16-04585-f004:**
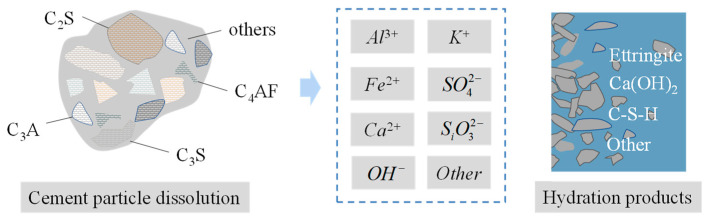
Cement hydration mechanism.

**Figure 5 materials-16-04585-f005:**
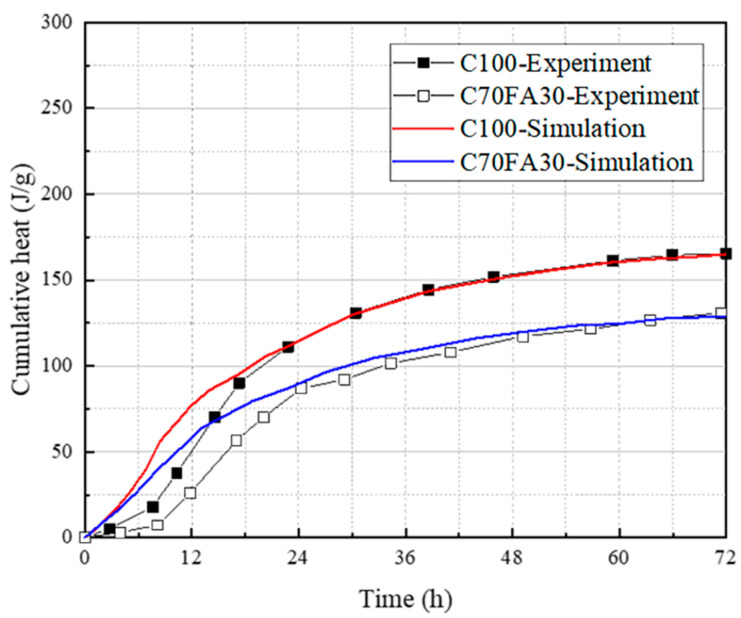
Simulation results of cumulative heat due to the effect of fly ash.

**Figure 6 materials-16-04585-f006:**
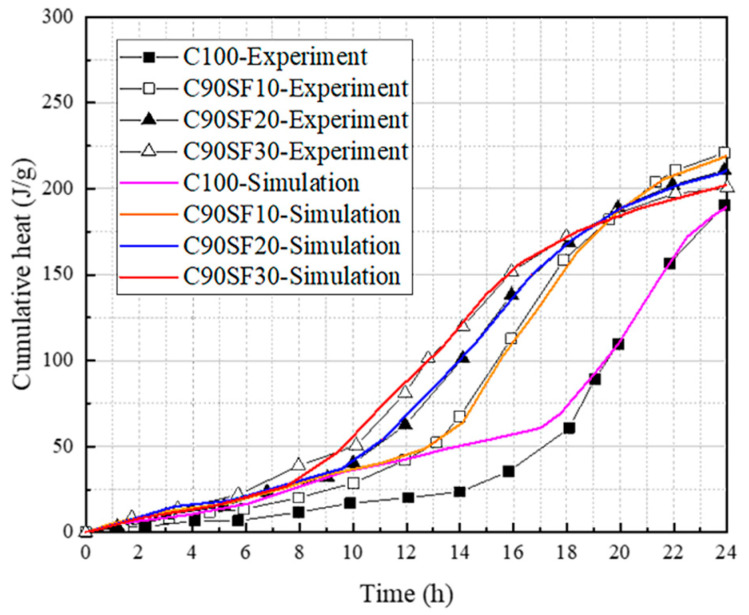
Simulation results of cumulative heat due to the effect of silica fume.

**Figure 7 materials-16-04585-f007:**
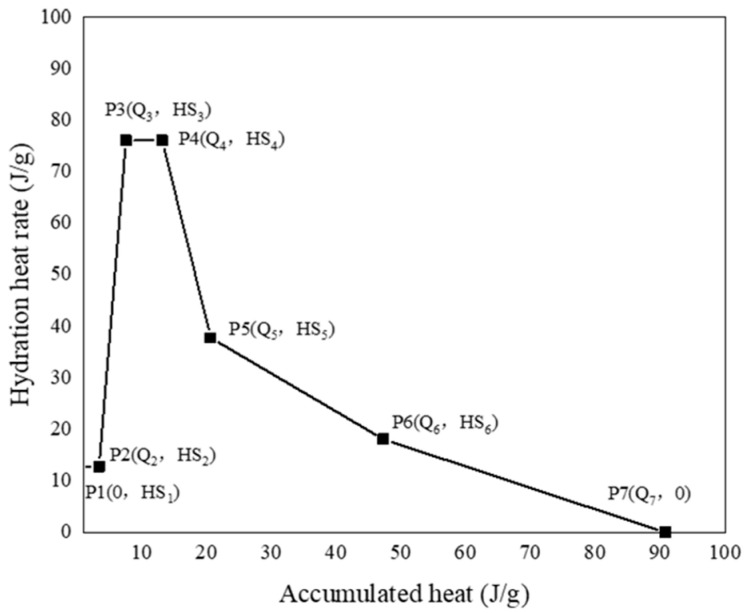
Exothermic process of the micro-filling effect of limestone powder.

**Figure 8 materials-16-04585-f008:**
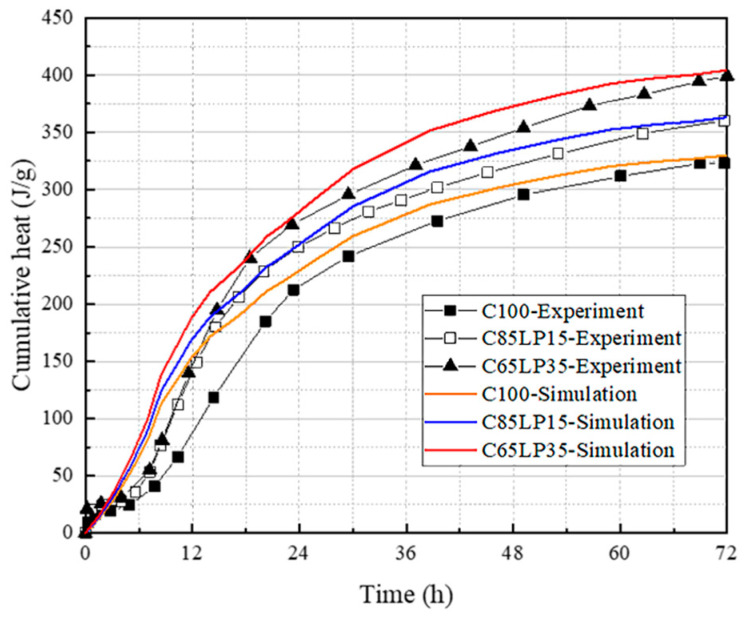
Simulation results of cumulative heat due to the effect of limestone powder.

**Figure 9 materials-16-04585-f009:**
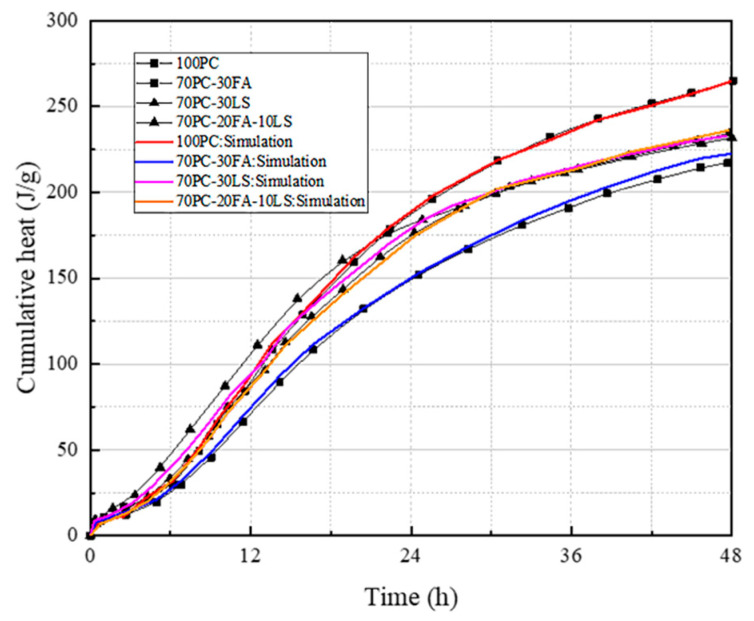
Simulation results of cumulative heat due to the effect of the multi-component powders.

**Figure 10 materials-16-04585-f010:**
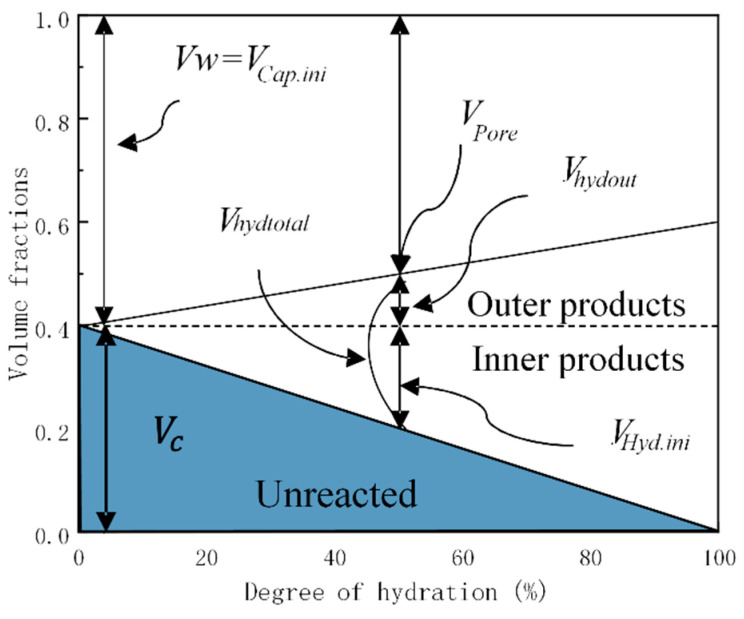
Changes in the volume of the components and their indicator for strength development.

**Figure 11 materials-16-04585-f011:**
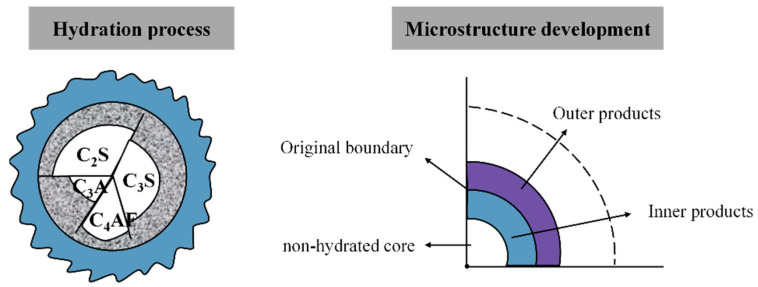
Hydration and development of cement microstructure.

**Figure 12 materials-16-04585-f012:**
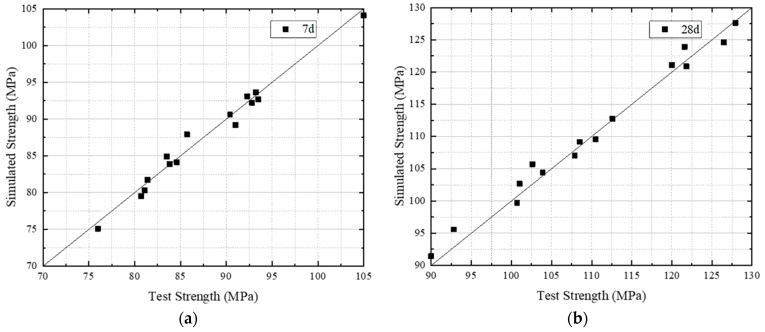
Prediction results of compressive strength of different mix designs using the empirical model. (**a**) The comparison between 7 d strength prediction and test results. (**b**) The comparison between 28 d strength prediction and test results.

**Figure 13 materials-16-04585-f013:**
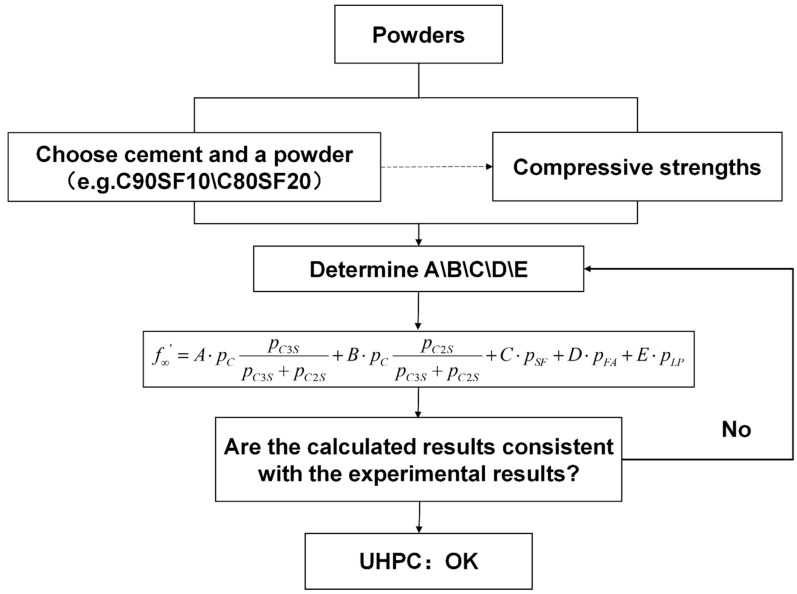
Simplified mix design process based on the strength prediction model.

**Figure 14 materials-16-04585-f014:**
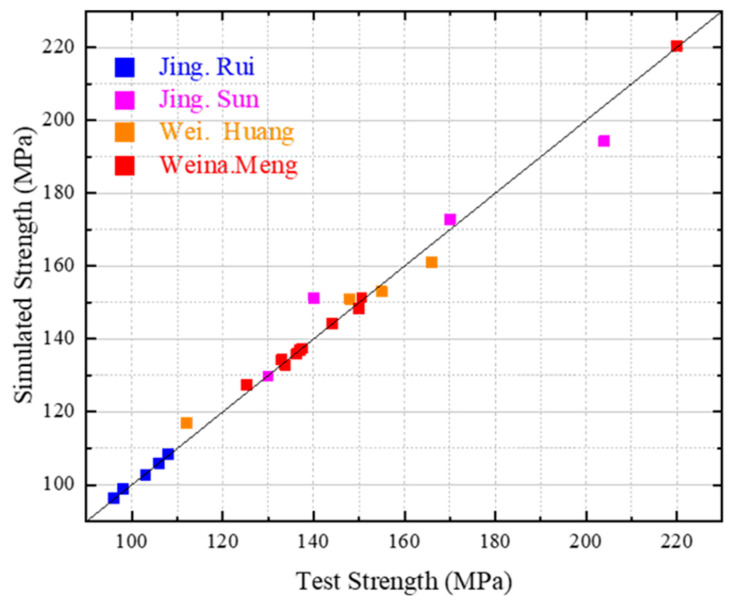
Verification of the applicability of the model based on the mixture design data of other references.

**Table 1 materials-16-04585-t001:** Chemical and mineralogical compositions of Portland cement.

Chemical Component	Content (wt.%)	Mineralogical Component	Content (wt.%)
CaO	64.51	C_3_S	69.71
SiO_2_	20.36	C_2_S	5.87
Fe_2_O_3_	2.82	C_3_A	5.73
Al_2_O_3_	3.96	C_4_AF	8.57
SO_3_	2.61	Gypsum	4.44
MgO	2.26	Arcanite	1.4
K_2_O	1.22		
TiO_2_	0.496		
BaO	0.379		
SrO	0.226		
Na_2_O	0.10		

**Table 2 materials-16-04585-t002:** Sample codes used for the investigated mix design (kg/m^3^).

Group	Code	Cement	Silica Fume	Fly Ash	Limestone Powder	Aggregate	Water	Superplasticizer	Steel Fiber
1	C100	1000	0	0	0	1100	180	30	157
2	C95SF5	950	50	0	0	1100	180	30	157
3	C90SF10	900	100	0	0	1100	180	30	157
4	C80SF20	800	200	0	0	1100	180	30	157
5	C60SF40	600	400	0	0	1100	180	30	157
6	C80SF10FA10	800	100	100	0	1100	180	30	157
7	C70SF10FA20	700	100	200	0	1100	180	30	157
8	C60SF10FA30	600	100	300	0	1100	180	30	157
9	C70SF20FA10	700	200	100	0	1100	180	30	157
10	C60FA40	600	0	400	0	1100	180	30	157
11	C80SF10LP10	800	100	0	100	1100	180	30	157
12	C70SF10LP20	700	100	0	200	1100	180	30	157
13	C60SF10LP30	600	100	0	300	1100	180	30	157
14	C70SF20LP10	700	200	0	100	1100	180	30	157
15	C60LP40	600	0	0	400	1100	180	30	157

**Table 3 materials-16-04585-t003:** Effect of mix design composition on compressive strength.

Group	Code	Compressive Strength (7 d)	Compressive Strength (28 d)	Slump Flow (mm)
1	C100	76	90	630
2	C95SF5	80.7	92.8	550
3	C90SF10	81.4	100.7	580
4	C80SF20	83.5	102.6	650
5	C60SF40	105	126.5	400
6	C80SF10FA10	85.7	103.9	730
7	C70SF10FA20	92.3	121.8	700
8	C60SF10FA30	93.2	127.9	610
9	C70SF20FA10	81.1	101	600
10	C60FA40	93.5	121.6	750
11	C80SF10LP10	91	101	650
12	C70SF10LP20	83.8	101.7	710
13	C60SF10LP30	84.6	105.6	730
14	C70SF20LP10	90.4	110.8	620
15	C60LP40	92.8	120	780

## Data Availability

Not applicable.

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
