# Peer review of "Empirical Compression Model of Ultra-High-Performance Concrete Considering the Effect of Cement Hydration on Particle Packing Characteristics"

_materials, 2023, doi:10.3390/ma16134585_

Round 1

Reviewer 1 Report

The similarity index of the manuscript is high. Please reduce the similarity to below 15% by rephrasing portions of text that have a high similarity to other sources.

There are many flaws in formatting throughout the manuscript. Please minimize the flaws by referring to the template of the journal and following the format presented in the template meticulously.

Citations are missing throughout the manuscript. Please add the citations.

The majority of the references are outdated. Please cite references that are more recent and are relevant to the study.

The gap in research that the study is attempting to fill is not clearly highlighted. Please highlight the gap clearly.

The novelty of the study is not clearly highlighted. Please highlight the novelty clearly.

There are too many figures throughout the manuscript. Please reduce the number of figures by presenting the figures that contain the key findings and discussing them in detail while removing figures that are not as important.

Recommendations for future research is missing. Please highlight the recommendations after concluding the key findings.

There are many flaws in the usage of English throughout the manuscript. Please minimize the flaws by consulting with a professional English proofreader.

Author Response

Special thanks to you for your good comments. The following is the response to your comments:

  1. The article has been re-checked and the parts with high repetition rate have been rewritten to reduce the similarity.
  2. I am very sorry for the lack of care in referencing the magazine template, and the formatting has now been standardized.
  3. Most of the citations in the full text are in the introduction section, and some citations are missing from the later analysis of the experimental results, which have now been added.
  4. The aim of this study is to develop a strength model that can predict the compressive strength of ultra-high-performance concrete to improve the stability of tunnel lining structures. Additional information has been provided in the last paragraph of the introduction.
  5. In this study, a UHPC compressive strength prediction model is designed by considering the chemical process of exothermic hydration based on the physical stacking state and using this as the basis for strength prediction using only a small amount of experimental data. It has now been added to line 447 in Conclusions.
  6. The charts in the full text have been revised and adjusted appropriately, and unimportant charts have been removed.
  7. For the shortcomings and outlook of this paper's research, they have been added in the fourth point of conclusion.
  8. The official language editing service has been performed with MDPI to correct the flaws in English usage throughout the text. A certificate attesting to the official language changes has been included at the end of the references.

Reviewer 2 Report

In their study, the authors analyze the properties of ultra-high-performance concrete (UHPC) under the influence of different contents of silica fume, fly ash, and limestone powder. They also developed models to predict heat release, pore development, and compressive strength. The prediction method proved accurate for different UHPC mix designs investigated here and may be of use to the people working in the area. In my opinion, the work is well written and can be accepted, after minor corrections related to the list of references which should be indicated in text in the increasing order. 

Author Response

Thank you very much for your approval of the research content of this paper and for your valuable comments. The references cited in the text have been revised and are shown in ascending order.

Reviewer 3 Report

In the present paper, Empirical compression model of ultra-high performance concrete considering the effect of cement hydration on particle packing characteristics. The article needs a major revision based on the following comments.

1.      The research introduction needs more attention than the current situation to show the gap between what has been studied and the importance of the current research.

2.      References needs to be updated up to last year.

3.      I think to improve the quality of the paper, the workability of all the mixes needs to be provided

4.      In lines 145 and 146, the authors stated that flexural strength were carried out and in the experimental results the authors did not discuss it, so it is necessary to explain and discuss the effect of silica fume (SF), fly ash (FA), limestone powder (LP)  on the flexural strength of UHPC.

5.      In scientific research subject pronouns are not used, so all pronouns must be deleted, such as (We).

6.      In lines 11 and 95 delete Ultra-high-performance concrete and use the abbreviation of UHPC.

7. To improve the paper's quality, a proposed formula to predict flexural strength should be developed.

8.      In the conclusion section in point 1, the author stated that the mechanical properties of UHPC with different components were investigated but only compressive strength was investigated, please discuss the other UHPC properties.

English Language must be revised

Author Response

Special thanks to you for your good comments. The following is the response to your comments:

  1. References have been updated to last year's literature, examples of references 38, 39, 41.
  2. Data for the expansion of different fit ratios have been added in Table 3 to characterize the work performance.
  3. Since the effect of compressive strength is mainly discussed in the whole paper, the effect of different powders on flexural strength is not specifically analyzed. Also the proposed formula for predicting the flexural strength will be reflected in the outlook and will be continued in subsequent studies.
  4. The entire text has been combed and all subject pronouns have been removed, while all references to ultra-high-performance concrete, except for the abstract, have been replaced with the abbreviation UHPC.

Reviewer 4 Report

The manuscript contains many experimental and analytical results, but needs much improvement in terms of scientific and English writing. Specifically, the authors explain too much at length. This is counterproductive. The current manuscript is written so that only the title and abstract are readable.

In addition, the biggest challenge in research related to the development of predictive models is the reliability of the model. It is difficult for reviewers and readers to trust the developed model based only on the manuscript. Consider this and revise the manuscript conservatively.

To be honest, even the reviewer found this manuscript difficult to read for the above reasons (especially the introduction and discussion). Thus, the manuscript should be revised to at least be readable and understandable by the reviewer.

Here are some additional comments

- The reference style is incorrect, and even the reference numbers are not sequential. Please review and revise the entire manuscript.

- In the introduction, it is problematic to state that “there is no existing theoretical basis or model”. The authors can never read all the existing literature. Revise conservatively.

- In Section 2, the units for specific surface area are strange. In addition, the properties of the raw materials (specific surface area, density, particle size, etc.) should be organized in a table.

- In Table 1, the chemical and mineralogical compositions of Portland cement were determined by what equipment and process? Specify.  

- In Table 2, specify the units. Also specify the mix proportions of all raw materials used (including aggregates, water-reducing agents, steel fibers, etc.).

- On line 143, the parentheses are incorrectly formatted. Check.  

- Conclusions can't be all-encompassing. Keep it concise and focused on the key findings of your paper.

The manuscript contains many experimental and analytical results, but needs much improvement in terms of scientific and English writing. 

Author Response

Special thanks to you for your good comments. I am sorry that the flawed English usage caused you to have trouble reading, but the English grammar has now been corrected throughout the text by the official MDPI language editor. The following is the response to your comments:

  1. The cited references have been uniformly numbered and the format of the references has been revised.
  2. After reviewing the relevant literature, the unit of specific surface area can be expressed in m2/kg, which is consistent with this paper therefore not much modification was done. Also, in order to reduce the number of diagrams in the article, the properties of the raw materials were not organized in a single table, but were described in a textual narrative.
  3. The chemical and mineral composition of Portland cement is determined by X-ray fluorescence spectroscopy (XRF) tests, and on this basis the mineral composition ratios are calculated by the Bogue method.
  4. Additional descriptions of the raw material mix proportions for all the mixes have been provided in Table 2, and the conclusions of the paper have been streamlined in an effort to highlight the key findings of the paper.

Round 2

Reviewer 1 Report

GENERAL

Correct the format of citations

Check all citations and references, as some of them are not cited in the text, and address any issue accordingly

Replace outdated references, where applicable, with those that were published since 2018

ABSTRACT

Line 9: Prior to stating the aim of the study, concisely highlight the problem that the study is attempting to address

Line 19: Objectively and concisely highlight the accuracy and precision of the prediction method

INTRODUCTION

Line 69: Introduce SF as the abbreviation of silica fume and then use SF onwards

MATERIALS AND METHODS

Move all discussion of results to a subsequent section as the Materials and Methods section is focused on describing the materials employed and methods adopted in the study

RESULTS AND DISCUSSION

Re-structure the sections that discuss the results in order to move all results that are reported in the Materials and Methods section to the appropriate sections

CONCLUSIONS

Provide recommendations for future research based on the findings of the present study

Proofread the manuscript with an English language professional after performing the revisions as, although the quality of English has greatly improved throughout the manuscript in relation to the previous draft, several minor flaws in the usage of English were present and have to be addressed

Reviewer 3 Report

paper can be accepted in the present form

Author Response

Many thanks for your approval.

Reviewer 4 Report

Line 92-97: The last paragraph of the introduction should be about the purpose of the study, but the current one does not match the title of the study. Thus the last paragraph should be revised to be consistent with the overall concept and title of the study.

Line 397-407: I don't understand why the purpose of the study appears in this paragraph. It should be moved to the last paragraph of the introduction.

none.

Author Response

Thank you very much for your valuable advice. The logical deficiencies in the context of the text have been corrected and revised according to your suggestion.